

# Modelling reference evapotranspiration of green walls ($ET_0^{vert}$)

Karin A. Hoffmann[1], Rabea Saad[2], Björn Kluge[1], Thomas Nehls[1*]

[1]Chair of Ecohydrology and Landscape Evaluation, Technische Universität Berlin, 10587 Berlin, Germany
[2] Fachbereich Umwelt, Stadt Braunschweig, Willy-Brandt-Platz 13, 38102 Braunschweig, Germany

*Correspondence to*: Thomas Nehls (thomas.nehls@tu-berlin.de)

**Short Summary**

$ET_0^{vert}$ is a process-based model concept for evapotranspiration of green walls, validated with onsite lysimetry data. Best accuracy of predictions is achieved using input data measured onsite and considering height dependencies of radiation, wind and VPD. If only remote climate station data is available as input, it must be "verticalized". The model predicts the hourly and daily evapotranspiration necessary for e.g., irrigation planning, building energy simulations or local climate modeling.





**Abstract**

Green walls, façade greenery, living walls – vertical building greening as part of urban green infrastructure are measures for climate sensitive urban design, for water management and microclimate regulation. Strategic integration of green walls into local water and energy cycles requires prediction of evapotranspiration, considering the individual design, plant species, and building characteristics. Available models address horizontal surfaces but disregard vertical particularities and urban conditions, e.g., reduced direct radiation, spatial patterns of radiation on the wall due to building orientation and shading

obstacles, and very heterogeneous wind fields that are influenced by rough surfaces, canyons, and adjacent wind barriers. We present a verticalization model, $ET_0^{vert}$, for the reference crop evapotranspiration $ET_0$ (FAO) based on a sensitivity analysis. It comprises the adaptation of solar radiation and wind to the individual situations in front of a wall or facade. The accuracies of the model predictions are evaluated for (i) remote climate station data (horizontal reference plane), (ii) interpolated climate data (both horizontal and vertical reference plane) and (iii) on-site measured climate data (vertical

reference plane, both not height-adapted and height-adapted) as input. We validate the model with data for a one-month reference period (25/07/2014 – 29/08/2014) from a weighable lysimeter with *Fallopia baldschuanica* greening of a 12 m high wall in Berlin, Germany.

Regarding individual meteorological input parameters, we found high relevance of both vapor pressure deficit (*VPD*) and solar radiation ($R_S$) for the study area. Using *VPD* and $R_S$, respectively, a linear model could explain 90 % and 85 % of daily

$ET_0$ variances. No such relationship could be detected for wind speed, but for maximum and minimum wind speed. Compared to remote horizontal input data, verticalization of input data ($R_S$ and wind) reduced overestimations of *ET* from about 90% to 14% and 27% for the daily and hourly resolution, respectively. If onsite climate data is available, deviations are reduced to 9% and 5% for the daily and hourly resolution. Height-adaptation of input data resulted in further improvements of the prediction accuracies (1% and 2% deviation for hourly and daily resolution).

We conclude that simply using remote horizontal climate data for calculating *ET* of green walls is not advisable. Instead, any input data, onsite measured or remote climate station data, should be verticalized and preferably height-adapted.
The verticalized model predicts the hourly and daily evapotranspiration of green walls necessary for e.g., irrigation planning, building energy simulations or local climate modeling.

## 1 Introduction

In cities the urban heat island effect amplifies temperature effects of climate change (McCarthy et al., 2010; Oke et al., 2017), confronting urban populations with heat stress risks (Buchin et al., 2016). Vertical green can be used for enhancing thermal comfort on indoors and outdoors (Hoelscher et al, 2016, Koch et al., 2020). Despite research quantifying the cooling effects of vertical green, process-based models predicting its evapotranspiration (*ET*) and therefore its water demand are not available (Bartesaghi-Koc et al., 2018) and therefore addressed in this article.

Vertical greenery systems (VGS) are understood as a general term for different types of green facades (i.e., ground-based, wall-based and pot-based) (Langergraber et al., 2021). Their benefits include reduction of temperature extremes in summer indoor and outdoor (Koch et al., 2020, Hoffmann et al., 2021) through isothermic shading and evaporative cooling (Wong et al., 2010, Hoelscher et al, 2016, Perini et al., 2017), thermal insulation due to reduction of wind speed (Perini et al., 2011), acoustic insulation (Azkorra et al., 2015), synergies with photovoltaic systems (Penaranda Moren and Korjenic, 2017),

deposition of particulate matter on the leaves (Ottelé et al., 2010), carbon sequestration (Charoenkit and Yiemwattana, 2016), food production (Hoffmann et al., 2023), creation of fauna habitats (Madre et al., 2015) and improvement of the human wellbeing and environmental justice (Felgentreff et al., 2022). Two other important benefits, which are both closely linked to the plants' *ET*, are rainwater retention (Riechel et al., 2017; Xie et al., 2022), and rainwater management (Pearlmutter et al., 2021).





Regarding the cooling effects, Hoelscher et al. (2016) investigated the contribution of shading, transpiration, and insulation processes. They measured daily transpiration contributions of 13 % up to 73 % depending on species and on weather conditions. Cameron et al. (2014) found the contribution of transpiration to vary from 18 % to 55 %. Hoffmann et al. (2021) simulated the cooling performance of a VGS for a set of nine buildings of different wall compositions, i.e., materials with specific thermal transmittance and thermal inertia, and southern and western wall orientations. They found highest reductions

of energy inflow into the building for a commercial hall and a prefabricated panel building. However, studies calculating the evapotranspiration of VGS are scarce. This knowledge gap entails uncertainties quantifying the cooling effects of VGS, regarding potentials of rainwater management as well as irrigation requirements.

*Measuring ET of VGS*

Studies measuring the transpiration of VGS are scarce as well (Stec et al., 2005; Hoelscher et al., 2018; Hunter et al., 2014;

Salonen et al., 2022). First insights about the $ET$ of VGS can be obtained by looking at the irrigation water demands of wall bound VGS with small buffer volumes. These differ strongly between locations, VGS types, plant species, substrates, and seasons (Pérez-Urrestarazu and Urrestarazu, 2018). In summer, green walls were found to have a demand of 2.5–5 $L\,m^{-2}\,d^{-1}$, while in fall and winter it drops to 0.5–1.8 $L\,m^{-2}\,d^{-1}$ (Fernández Cañero et al., 2011; Perini et al., 2017; Prodanovic et al., 2019). Few studies so far have reported direct measurements of VGS transpiration. Borowski et al. (2009) investigated

photosynthetic rates and water use efficiencies of three climber species in urban surroundings using a potometer. They report of maximum transpiration rates for fully sun-exposed leaves of *Hedera helix* of up to 4 $L\,m^{-2}\,d^{-1}$, for *Fallopia baldschuanica* of up to 7.7 $L\,m^{-2}\,d^{-1}$ and for *Vitis riparia* of up to 3.3 $L\,m^{-2}\,d^{-1}$. Leuzinger et al. (2011) also conducted potometer measurements on *Hedera helix*, supplemented by sap flow measurements, albeit in forest surroundings. They observed much smaller transpiration rates averaged across all leaves (0.36 $L\,m^{-2}\,d^{-1}$) than maximum leaf transpiration under conditions of full

irradiation (2.9 $L\,m^{-2}\,d^{-1}$).

Hoelscher et al. (2016, 2018) determined $ET$ from sap flow and roofed lysimeter measurements of three climbing plant species on urban facades during August and September. Here, *Fallopia baldschuanica* transpired between 0.7–2.3 $L\,m^{-2}\,d^{-1}$, *Hedera helix 1.2*–1.7 $L\,m^{-2}\,d^{-1}$ and *Parthenocissus tricuspidata* 0.7–1.3 $L\,m^{-2}\,d^{-1}$. Wolter (2015) measured the water demand of a living wall panel system planted with *Hedera helix* for the period of May to November and found the daily water

demand ranging between 0.1–6.8 $L\,m^{-2}\,d^{-1}$. Both refer to the vertical wall area. The panel system exhibited a very high wall leaf area index of up to 7.75. The experimental design did not allow distinguishing between transpiration and evaporation from the substrate. Van de Wouw et al. (2017) monitored the water flows and mass changes of two living wall systems, a planter box, and a panel system, both planted with a mix of predominantly wintergreen species. The water use in the test period in winter was 0.81 and 0.75 $L\,m^{-2}\,d^{-1}$, respectively. Measuring $ET$ is challenging regarding setup, time, and financial

resources while results cannot be generalized at least because of the site-specific conditions. In turn, calculating $ET$ specifically with process-based models for standardized conditions is a well-established reference method to derive information for irrigation planning (Allen et al., 1998). Therefore, adapting the $ET_0$ concept for VGS would be advantageous.

*Modeling ET$_0$ of VGS*

Studies modeling the transpiration of VGS used empirical or physically based approaches (e.g., Malys et al., 2014; Stec et

al., 2005; van de Wouw et al., 2017) with models based on the Penman-Monteith equation (Allen et al., 1998) delivering most accurate estimates for $ET$. Stec et al. (2005) used the Penman-Monteith equation as part of their heat-exchange model for a double-skin facade with plants between the glass panel and the wall. They used the surface temperature of the plants' leaves measured in a controlled laboratory environment for validation. Although that revealed a near-perfect correlation





($R^2 = 0.98$), one must criticize, that $ET$ was not individually measured. Malys et al. (2014) also included the Penman-

Monteith equation and calculated $ET$ values. They correlated the values to surface temperatures and to $ET$ derived from mass changes of the investigated green wall samples. While leaf surface temperatures also correlated well between simulation and reality, $ET$ was profoundly underestimated, especially during peaks of high $ET$. So, the surface temperature alone is just a proxy for $ET$. Wolter (2015) measured mass changes of a green wall element to validate the Penman-Monteith estimates ($ET$(lysi)=0.76 $ET_p$ + 0.58, $R^2 = 0.72$) for daily values on a southern façade. Van de Wouw et al. (2017) investigated two

different types of VGS and concluded that coefficients of determination with estimated $ET$ values were much better for the panel system ($R^2 = 0.7$) than for the planter box system ($R^2 = 0.3$). Crop coefficients of 1.46 and 0.76 were proposed for panel and planter box systems, respectively (van de Wouw et al., 2017). Much smaller crop coefficients were calculated by Segovia-Cardozo et al. (2019) for their green wall (0.32 for winter and 0.6 for summer). Hoelscher (2018) calculated the $ET_0$ (FAO) and compared it to lysimeter measurements at a facade greened with *Fallopia baldschuanica*. A crop coefficient of

1.25 was determined. On a daily resolution, model uncertainty was quite high ($R^2 = 0.5$). However, many studies showed no validation of their estimation results (e.g., Yazdanseta and Norford, 2017) or modified the Penman-Monteith equation so drastically to represent their specific VGS, that their results lack transferability and comparability (Davis and Hirmer, 2015). In the latter case, radiation input was neglected as evapotranspiration was calculated in the air gap between substrate of a VGS and the wall which was shielded from radiation.

The estimation of $ET_0$ from climate data works best when these are measured onsite. When no onsite data is available, data from nearby stations can be taken as an approximation in homogenous landscapes (ASCE-EWRI, 2005; McMahon et al., 2013). Regarding urban sites and buildings, there are some limitations to the application of Penman-Monteith equation to VGS: (i) it was developed for agricultural crops. By the means of crop coefficients, the estimates can be scaled up to other plant stocks, such as meadows or forests. (ii) For isolated plants though, some studies have revealed uncertainties (e.g.,

Dragoni et al. (2006) for grapevines and Rana et al. (2019) for street trees). The Penman-Monteith model assumes homogenous horizontal areas of crops, so some assumptions concerning (iii) radiation budgets, (iv) wind speed patterns, (v) temperature and (vi) relative humidity patterns do not apply to vertical surfaces in cities. This includes horizontal and vertical variability caused by height, exposition, shading by surrounding buildings or even trees in front of the VGS, height-dependent sky view factors and small-scale climatic effects typical for urban areas (Gong et al., 2018) as the urban matrix

causes strong temporal and spatial climatic heterogeneity (Arnfield, 2003).

The availability of solar radiation in cities depends on the diverse urban morphology and the associated differences in the diurnal cycles of direct solar radiation (Arnfield, 1990; Littlefair, 2001). The derivation of radiation availability for a certain point in time and space from city surface models has been a focus in urban climate research. Many radiation distribution models for urban canyons exist, ranging from empirical approaches to more sophisticated 3D analyses and featuring

different levels of spatial and temporal resolution (Freitas et al., 2015). The basic principle of these models is the geometrical computation of skyline obstructions (Arnfield, 2003, Lindberg et al., 2015, 2018) resulting in a diurnal cycle of direct irradiation, which is then used to scale radiation measured at an unobstructed point nearby. For example, the RayMan model can simulate time series of radiation flux densities from a digital elevation model (Matzarakis et al., 2007, 2010). It is a freely available, widely used, and validated tool (e.g., Hämmerle et al., 2011; Jänicke et al., 2015; Gál and Kántor, 2020) and

has already been used for $ET$ calculations (Paparrizos et al., 2014).

Wind speed profiles in urban areas differ strongly from those over open fields due to the increased surface roughness and show high variations due to specific local conditions and obstacles (Arnfield, 2003; Barlow and Drew, 2015). Nakamura and Oke (1988) investigated the link between wind speeds in an urban canyon and above roof-height and found a linear relationship which was the stronger the higher the wind speeds were. The slope of this linear relationship depends on the





canyon geometry (especially height-to-width ratio) and the position of measurements in and above the canyon (Nakamura and Oke, 1988). To refine that estimation, wind direction in relation to the canyon long axis can be included in the model. Buller (1976) first showed the general tendency of the factor describing the relationship between in-canyon and above-canyon wind speeds to increase with the above-canyon wind direction shifting from perpendicular to parallel to the canyon long axis. More recent studies (Christen, 2005; Eliasson et al., 2006) confirmed the strong dependency of wind speed profiles in an urban canyon on the wind direction above the canyon. Instead of a linear relation between wind speed and height, the wind profile can also be assumed to be exponential up to roof height and logarithmic above (Barlow and Drew, 2015). Numerical simulations based on computational fluid dynamics (CFD) have been used to model wind flow in urban areas in high spatial and temporal resolution (Murakami et al., 1999; Di Sabatino et al., 2013).

Air temperature in urban areas is generally higher compared to surrounding rural sites - the well described Urban Heat Island (UHI) effect (Arnfield, 2003; Oke, 1976, 1982). Also, there is considerable intra-urban spatial heterogeneity in air temperatures linked to differences in land use, surface cover, sky view factors and vegetation (Eliasson and Svensson, 2003; Unger, 2004; Fenner et al., 2014).

Air humidity variations between urban and rural areas have been thoroughly investigated (e.g., Ackerman, 1987; Fortuniak et al., 2006; Lokoshchenko, 2017), while only few studies so far have investigated intra-urban variation in humidity (e.g., Henry et al., 1985; Wang et al., 2017; Yang et al., 2017). While many studies found humidity decreased at urban sites, coined "Urban Dry Island" (e.g., Ackerman, 1987; Cuadrat et al., 2015; Lokoshchenko, 2017), some studies identified an urban moisture excess (Kuttler et al., 2007; Unger, 1999). Intra-urban heterogeneity in humidity was linked to land use and surface materials (Henry et al., 1985), impervious surface area fraction (Wang et al., 2017), topography, urban density, and vegetation abundance, anthropogenic humidity (Cuadrat et al., 2015; Yang et al., 2017).

*Aims of the study*

Our research aimed to (i) introduce a model adaptation for the daily/hourly reference evapotranspiration $ET_0$ of a VGS called $ET_0^{vert}$ e.g. for irrigation planning or cooling potential quantification, requiring only local climate data as well as information on the city morphology at the site e.g., from digital elevation models. Based on (ii) a sensitivity analysis the most relevant parameters influencing $ET_0$ (ASCE-EWRI, 2005) were adapted to the micro-climatic conditions at a specific urban site and (iii) a validation of the model was performed using vertical green lysimeter data and onsite measurements of climatic conditions on a VGS stand in Berlin, Germany.

**2 Theory**

Most available model concepts have been developed for horizontal crop stands. They need to be adapted from describing horizontal to vertical conditions to be applied for VGS (see Fig.1, Fig.2). Depending on the availability of input data and required accuracy, different approaches for the estimation of $ET$ are available. They span from regionally validated, empirical approaches (e.g., Haude, Hargreaves) for monthly to weekly resolution, e.g., Hargreaves' approach is recommended for time periods of more than five days (Hargreaves and Allen, 2003), to process-based approaches such as Penman-Monteith experimentally validated for down to 10-minute time steps (Fank, 2007). The widely accepted Penman-Monteith approach has been adapted for the calculation of the standard evapotranspiration $ET_0$ recommended by the Food and Agriculture Organization of the United Nations (FAO 56, Allen et al., 1998).

We have therefore chosen the $ET_0$ concept for process-based modelling of the evapotranspiration of VGS. In the following we will elaborate the theoretical concept of verticalization of $ET_0$ to $ET_0^{vert}$, that describes the evapotranspiration of a vertical-area increment of 1 m². It is planted with grass in a defined height at a wall situated e.g., in an urban street canyon.





Using the $ET_0^{vert}$ concept later allows the application of the crop coefficient ($K_C$) concept, with individual $K_C$ values for plant

species used in VGS: $ET_C^{vert} = K_C\ ET_0^{vert}$ .

$ET_0$ is sensitive to net solar radiation, $R_n$, and thus to incoming short-wave solar radiation $R_S$, wind speed $u$, air temperature $T$ and water vapor pressure deficit ($VPD = e_s - e_a$). Anticipating the sensitivity analysis, the sensitivity of $ET_0$ for these parameters is depicted in Fig. 3 for the variability of their values at nearby climate stations. Due to the changing sensitivity over the range of values, all these parameters need to be verticalized, so that:

$ET_0{^{vert}}_h = f(h) = ET_0(R_{n_h}^{vert}, u_h^{vert}, T_h^{vert}, (e_s - e_a)_h^{vert})$       (mm h⁻¹)       (eq. 1)

With $ET_0$ being calculated using the standardized form of the Penman-Monteith equation proposed by Allen et al. (2005) in the Task Committee on Standardization of Reference Evapotranspiration of the American Society of Civil Engineers (ASCE) for daily time steps and short surfaces:

$$ET_0 = \frac{0.408\ \Delta\ (R_n - G) + \gamma\ \frac{C_n}{T + 273}\ u_2\ (e_s - e_a)}{\Delta + \gamma\ (1 + C_d\ u_2)}$$       (mm h⁻¹)       (eq. 2)

with:

$ET_0$     standardized reference crop evapotranspiration for short surfaces (mm h⁻¹),

$R_n$     calculated net radiation at the crop surface (MJ m⁻² h⁻¹),

$G$     soil heat flux density at the soil surface (MJ m⁻² h⁻¹),

$T$     mean hourly air temperature at 1.5 to 2.5 m height (°C),

$u_2$     mean hourly wind speed at 2 m height (m s⁻¹),

$e_s$     saturation vapor pressure at 1.5 – 2.5 m height (kPa), calculated as the average of saturation vapor pressure at max. and min. air temperature,

$e_a$     mean actual vapor pressure at 1.5–2.5 m height (kPa),

$\Delta$     slope of the saturation vapor pressure-temperature curve (kPa °C⁻¹),

$\gamma$     psychrometric constant (kPa °C⁻¹),

$C_n$     numerator constant: for short surfaces and hourly timesteps equals 37 (K mm s³ Mg⁻¹ h⁻¹) and

$C_d$     denominator constant; for short surfaces and hourly data equals 0.24 during daytime and 0.96 during nighttime (s m⁻¹).

For $ET_0$ of a full VGS stand ($ET_0^{VGS}$), greater than 1 m², its vertical ($v$) and horizontal dimension ($h$) needs to be considered. The $ET_0{^{vert}}_h$ for different height increments must be summed to $ET_0^{VGS}$ according to:

$ET_0^{VGS}(h) = \sum_{i=0}^{h} (ET_0^{vert})_i$       (mm h⁻¹)       (eq. 3)

Therefore, the VGS stand height $h$ must be communicated (like the crown diameter for trees). $ET_0^{VGS}$ is relevant for practical irrigation planning and compatible and comparable to classical horizontal approaches relevant in spatial planning and hydrology. The importance of VGS as measures to increase latent heat flow in urban areas gets visible only in $ET_0^{VGS}$. In the following the verticalization approaches of the different parameters are described.

*Verticalization of solar radiation $R_S$*

The solar radiation is the primary energy source for the evapotranspiration process, and $ET_0$ is most sensitive to $R_S$, especially during the summer, when anthropogenic heat release is negligible (Pigeon et al., 2007). $R_s$ depends on the site of the building (height a.s.l., latitude), the exposition of the corresponding surface as well as altitude and azimuth of the sun, thus hour of the day and day of year. Weather conditions, mainly cloudiness and the site-specific air quality, both influencing

the turbidity, are also of great relevance. More important, in urban settings, the greened wall of interest might be surrounded by objects, such as buildings and trees, that are obscuring parts of its hemispheric radiating environment, expressed by a reduced sky view factor (SVF). Usually, higher wall increments receive more solar radiation than wall increments at the bottom of the street canyon. Therefore, the incoming shortwave radiation must be simulated for every wall increment, in every time step.





*Wall heat flux*

A second term in the energy balance is the soil heat flux $G$. It gets equivalent to the wall heat flux in $ET_0^{vert}$. In temperate regions, during the summer, the building becomes an energy sink while in parts of the winter, spring and autumn, energy from heating is transferred from the building into the VGS. During the vegetation period this could be of relevance and

should be accounted for in $G$. For the summer, Hoffmann (2019) estimated the peak wall heat flux into the building and out of the building to be around 20 W m$^{-2}$ while the incoming short-wave solar radiation was 450 W m$^{-2}$ for the west-oriented greened wall described by Hoelscher et al (2018). Therefore, when calculating $ET_0^{vert}$ as a measure for the plants' water demand, $G$ might be negligible at a daily base and during the day. During night however, $G$ gets negative. As measurements for $G$ were missing, the term ($R_n$ - $G$) was set to zero for negative differences as otherwise $ET_0$ might get negative. A

negative $ET_0$ could be interpreted as condensation. However, the formula was not derived and validated for condensation.

*Temperature and relative humidity*

$T$ and $rH$ combined in form of vapor pressure deficit (*VPD*) have a high impact on $ET_0$. The height dependencies of *VPD* depend on wind conditions and are much more complex than those of wind as demonstrated for $T$ by Offerle et al. (2007),

where $T$ in the street canyon did not show height dependencies apart from a period around noon. Only measurements at the site or complex CFD simulations would offer height-resolved input data on *VPD* for the calculation of $ET_0^{vert}$ and were therefore omitted in this study.

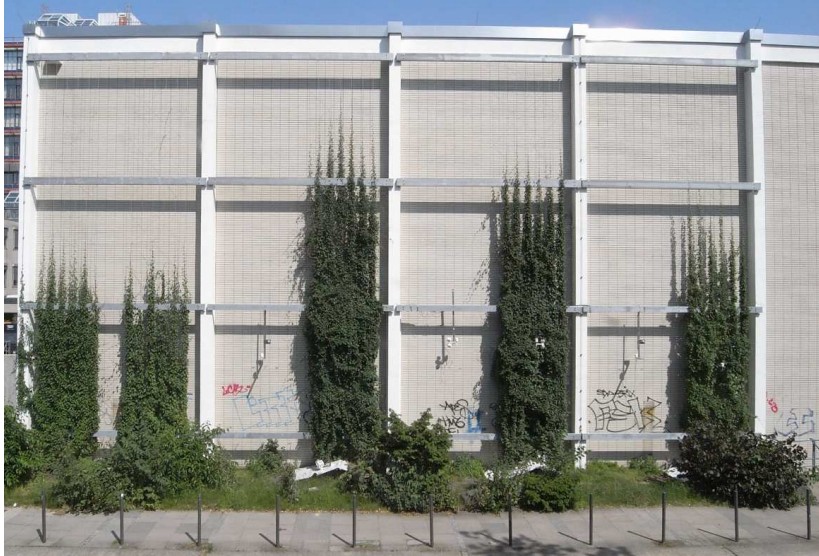

**Figure 1:** Vertical greening monitoring site in Berlin, Germany (western orientation), photographed at 6[th] August 2014

*Wind speed*

Wind speed in urban street canyons is known to depend on height and wind direction in relation to street canyon orientation. Eliasson et al. (2006) found a linear relationship of wind speed and height in the street canyon. Once, such linear

relationships are known for the site, they can be used to calculate a vertical wind profile which can be applied to calculate $ET_0^{vert}$. In $ET_0$ calculation, a standard height of 2 m is assumed. When wind data is available for heights other than 2 m, it can be converted using the following formula:



$$u_2 = u_z \frac{4.87}{\ln(67.8\,z - 5.42)} \qquad \text{(m s}^{-1}) \qquad \text{(eq. 4)}$$

with:

$u_2$     wind speed at 2 m above ground surface (m s$^{-1}$),

$z$     height of measurement above ground surface (m).


In conclusion, a whole set of climate data needs to be verticalized to calculate valid evapotranspiration for VGS. The
different expected impacts of the climate data on *ET* is depicted in figure 2.

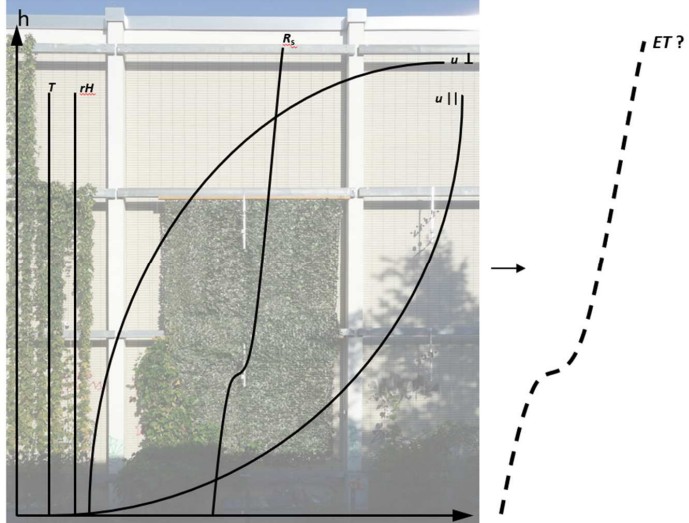

**Figure 2:** Expected height dependency of evapotranspiration *ET* because of height dependencies of shortwave radiation on vertical plane
($R_s$), wind speed orthogonal ($u\perp$) and parallel ($u\parallel$) to the facade. Note that temperature *T* and relative humidity *rH* are assumed to be
independent from height. The photo in the background depicts the 12 m high experimental vertical greening site in Berlin, Germany, with
shadows from neighbouring tree and building.

## 3 Material and Methods

### 245    3.1 Study site, experimental setup, and onsite measured data

The study site is situated in Berlin, Germany with a temperate oceanic climate (Cfb) according to the Koeppen climate
classification (Koeppen, 1936). The investigated year 2014 was one of the warmest years since the start of routine
temperature measurements in 1881. The average air temperature in Berlin was 11.3 °C, average precipitation sum was 470
mm and sunshine duration was 1719 h compared to 9.1 °C, 573 mm and 1635 h of sunshine hours for the reference period
1961–1990 (DWD, 2022). The study site (52.5136 N, 13.3243 E; 33 m a.s.l.) is a manufacturing hall located at the campus of
Technische Universität Berlin. The surroundings are characterized by dense midrise buildings among impervious land cover.
At the western wall (12 m height and 24.5 m width, see figure 1) of the hall five groups of six individuals each of *Fallopia
baldschuanica* were planted in containers. The plants were supported by climbing aids positioned 0.3 m in front of the wall.
The water table in the substrate was kept at a constant level by an automatic pump, which replenished the water from a
reservoir standing on a balance. The plant containers as well as the water reservoir were roofed to prevent rainwater
infiltration and minimize evaporation. It was measured and found to be negligible in relation to transpiration rates
(Schwarzer, 2015, Hoelscher et al., 2018). The facade area covered by the VGS, was estimated to be 18 m$^2$, 1.5 m broad and
12 m high, from time-laps photographs for the investigated period. The wall leaf area index of *F. baldschuanica* was





determined at the end of the growth period in 2014 to be 3.0 m$^2$ m$^{-2}$ (Hoelscher, 2018). Transpiration of *F. baldschuanica*

was determined by weighing the mass differences of the water reservoir (Signum1, Sartorius Weighing Technology GmbH, Germany) with a resolution of 2 g min$^{-1}$. According to Hoelscher et al. (2018), this resulted in an overall accuracy of 0.01 L m$^{-2}$. Negative differences have been aggregated to hourly and daily transpiration rates (L h$^{-1}$, L d$^{-1}$, respectively mm h$^{-1}$, mm d$^{-1}$), further denominated as $ET$(lysi). In addition, wind speed $u$ was measured in 3 m, 6 m, and 9 m height (Ultrasonic anemometer 3D, Adolf Thies GmbH & Co. KG), and relative humidity $rH$ (%, HC2-S3, Rotronic), air temperature $T$ (°C,

HC2-S3, Rotronic), and solar radiation $R_s$ (W m$^{-2}$, RA01-05 radiometer, Hukseflux) were measured in 3 m height. Both $u$, $T$, $rH$ and $R_s$ were measured at the central column of the wall, thereby $R_s$ was measured for the vertical plane. The horizon was mapped using a cherry picker and a theodolite (Theo 010, Carl Zeiss Jena, GDR) raised to 3 m height.

### 3.2 Sensitivity analysis of the Penman-Monteith equation

To investigate the impact of the meteorological variables on daily $ET_0$ for the respective study region hourly data from the

DWD station Berlin-Tegel, TXL (lat 52.5644 N, long 13.3088 E, 36 m a.s.l.) has been used. All relevant data is available for the years 2012–2020. The impacts of net radiation, water vapor pressure deficit and windspeed has been characterized by Pearson's correlation coefficient for the linear regression between the respective parameters and $ET_0$ for the horizontal case. The analysis has been repeated with data from the DWD station Potsdam (lat 52.3812 N, long 13.0622 E, 82 m a.s.l.) as here data for 30 years is available (1992–2021).

### 3.3 Remote and derived meteorological input data, and the according verticalization approaches for $ET_0$

As a remote data set, we used climate data ($T$, $rH$, $R_s$, $u$) provided by German Weather Service (DWD, 2022) for the stations airfield Berlin-Tegel (TXL), lat 52.5644 N, long 13.3088 E, 36 m a.s.l.) -hereafter called and labeled "(remote_TXL)" - and Potsdam (52.3812 N, long 13.0622 E, 82 m a.s.l.). Global radiation was measured for the horizontal plane as standard. Additionally, we used a data set from Meteonorm database (Meteotest, 2020) spatially interpolated to Berlin (lat 52.517 N,

long 13.3889 E, 43 m a.s.l.) for the year 2014. The radiation data was interpolated from the DWD stations Potsdam (26 km), Lindenberg (60 km) and Seehausen (119 km). Temperature data was interpolated from Berlin-Tempelhof (5 km), Potsdam (26 km), Neuruppin (85 km), Lindenberg (60 km), Slubice (84 km), and Holzdorf (85 km). $R_s$(remote_MNh) refers to the horizontal plane, meaning standard data, whereas in $R_s$(remote_MNv) refers to the vertical plane (calculated according to Perez et al., 1991) with the study site azimuth being 84 °. Time series were synchronized to UTC and full hour values refer to

the subsequent 60 minutes (e.g., 14:00 = 14:00–14:59).

To adjust $R_s$ for the different heights at the study site, the RayMan model was employed (Matzarakis et al., 2007, 2010) using spatial data from a digital surface model DOM1 (Geoportal Berlin, 2018) with 1 m spatial resolution as input. Radiation was simulated for the time series 25/07 – 29/08 for a vertical strip of 12 m$^2$ which corresponds to the position of radiation measurements (in 3 m height) at the study site.

Wind speed $u$ was approximated to different heights from a single measuring location on the wall. The height-dependent factors $k_u$ (for the heights 3, 6 and 9 m) were derived from onsite measurements in these three heights (see 3.1) to which a logarithmic model was fitted. Temperature and relative humidity were not assumed to be height dependent.

To assess the applicability of remote vs. onsite measured data, $ET_0^{vert}$ was calculated for the following five approaches:
(i) $ET_0^{vert}$(onsite_height), where the eight dependencies of $T$ and $rH$ were denied (measured onsite in 3 m height) while the

measured values for $u=f(h)$ and $R_s=f(h)$ were re-calculated for different heights, $h$. Wind speed has been inter-/ extrapolated from a logarithmic model based on measured data in 3 m, 6 m and 9 m heights. Wind has been measured 0.15 m from the





canopy, therefore eq. 4 has been used to convert $u=f(h)$ equivalent to measurements in 2 m distance to the wall. The insolation was calculated for the 12 height increments (1 m each) using $R_S$ (onsite_uni) measured onsite in 3 m height which was scaled using a linear scaling quotient $\frac{R_{S,Ray}(h)}{R_{S,Ray}(3m)}$ based on RayMan simulations (Matzarakis et al., 2007, 2010):

$R_S(h) = R_S(onsite) * \frac{R_{S,Ray}(h)}{R_{S,Ray}(3m)}$          (eq. 5)

| | |
|---|---|
| $R_S(h)$ | calculated solar radiation in height h (MJ m$^{-2}$ h$^{-1}$), |
| $R_S(onsite)$ | measured solar radiation in 3 m height (MJ m$^{-2}$ h$^{-1}$), |
| $R_{S,Ray}(h)$ | solar radiation calculated with RayMan in height h (MJ m$^{-2}$ h$^{-1}$), |
| $R_{S,Ray}(3m)$ | solar radiation calculated with RayMan in 3m height (MJ m$^{-2}$ h$^{-1}$). |

The results for $ET_0^{vert}$(onsite_height) were then calculated as the sum of $ET$ from all height increments.

(ii) $ET_0^{vert}$(onsite_uni), using onsite data measured at 3 m height for 1 m² and scaled up to the full plant area (18 m²) by multiplying with 18 resulting in a simplified, non-height dependent, uniform version. Similar to $ET_0^{vert}$(onsite_height), eq. 4

has been used to convert $u=f(h)$ measured in 0.15 m distance to the wall equivalent to measurements in 2 m distance to the wall.

(iii) $ET_0^{vert}$(remote_MNv) where climate input data was derived from Meteonorm database (Meteotest, 2020) using the mapped horizon on the site with interpolated data from near climate stations. $R_S$ refers to the vertical plane with an azimuth of 84° (Perez et al., 1991).

(iv) $ET_0^{vert}$(remote_MNh) where meteorological input data was derived from Meteonorm database (Meteotest, 2020) with interpolated data from near climate stations. $R_S$ refers to the horizontal plane.

(iv) $ET_0^{vert}$(remote_TXL) has been calculated with meteorological data from the DWD climate station Berlin Tegel.

The average model performance error is expressed by Mean Absolute Error (MAE) and Root Mean Square Error (RMSE).

The MAE is calculated as the average of the absolute errors with the formula

$MAE = \frac{1}{n}\sum_{i=1}^{n}|\dot{x}_i - x_i|$

where $n$ is the number of observations, $\dot{x}$ is calculated and x is observed data. RMSE is calculated with

$RMSE = \sqrt{\frac{1}{n}\sum_{i=1}^{n}(\dot{x}_i - x_i)^2}$

and is more sensitive to outliers than MAE. Both measures are 'dimensioned' and thus given in the unit of the evaluated

data.

### 4 Results & Discussion

#### 4.1 Sensitivity analysis of the $ET_0$ in the region

Regarding the above stated aims of this study, the model concept was formulated in section Theory (eq. 1), which based on the results of the following sensitivity analysis. For the station Berlin-Tegel (TXL), values for $ET_0$ range from 0 to

11.5 mm d$^{-1}$ (Fig. 3). Thereby, a linear model with $R_S$ explains 85 % of the variance of $ET_0$, for $VPD$ it is 90 % and $u$ 0.1 %. The relations are quite similar for the 30 years' time series of Potsdam climate station (distance to TXL is 26 km) with $R_S$



explaining 86 % of the variance of $ET_0$, $VPD$ explaining 92 % and $u$ explaining 1 %. Thus, $rH$ and $T$, respectively $VPD$ and solar radiation ($R_S$) are the most determining parameters for the corresponding region Berlin-Brandenburg, represented by climate stations airfield Berlin-Tegel (TXL) and a forested site in Potsdam-Telegrafenberg. This finding corresponds to the

general idea of the impact of energy and atmospheric water demand on $ET_0$. Changes in $u$ however influence changes in $ET_0$ to a small extent in the study region, as it is known for humid climates (Allen et al, 1998).

The sensitivity of $ET_0$ to the single climatic variables depends on location, as well as season (see Pérez-Urrestarazu and Urrestarazu, 2018). For the study region one can conclude, that $R_S$, $rH$ and $T$ must be adapted to the vertical site conditions while $u$ does not necessarily have to be adapted. As for other regions wind speed might have a higher impact, it will be

included in the following.

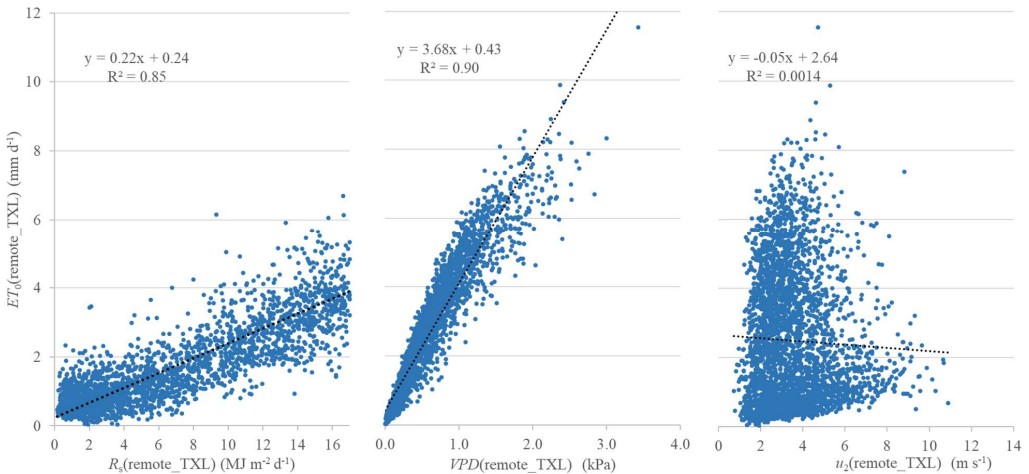

**Figure 3:** Relations between the input parameters solar radiation ($R_S$), vapor pressure deficit ($VPD$), wind speed measured in 2 m height ($u_2$) and standard reference evapotranspiration ($ET_0$) for daily resolution. Data originates from the climate station Berlin-Tegel (DWD, 2022) for the period 01/01/2012–31/12/2020; n = 3287.

### 4.2 Measured onsite versus remote climate station input data

Figure 4 shows parameters measured onsite ($R_S$, $VPD$, $u_3$, $u_6$, $u_9$) versus *data*(remote_TXL*)* and interpolated *data*(remote_MNh) and *data*(remote_MNv) for hourly and daily resolution.

Values for daily $R_s$(remote_MNv) overestimate daily $R_S$ (onsite_uni) by 26 % while overestimations are much higher using $R_S$ (remote_MNh) and $R_S$ (remote_TXL) with 133 % and 147 %, respectively. Values for $R_S$ (onsite_uni) correlate well with $R_S$ (remote_MNv) ($R^2$ = 0.96), $R_S$ (remote_TXL) ($R^2$ = 0.95) and $R_S$ (remote_MNh) ($R^2$ = 0.93).

For the hourly resolution, $R_S$ (onsite_uni) and $R_S$ (remote_MNv) ($R^2$ = 0.86) are in the same range whereas again, $R_S$ (remote_MNh) ($R^2$ = 0.56) and $R_S$ (remote_TXL) ($R^2$ = 0.53) show higher peaks and weaker correlation coefficients.

The good correlation between $R_S$ (onsite_uni) and $R_S$ (remote_MNv) could be expected for both temporal resolutions. The decrease in $R^2$ from daily to hourly resolution can be explained by weather dynamics like clouds passing the region. The low correlation between $R_S$ (onsite_uni) and the horizontal $R_S$ (remote_TXL) and $R_S$ (remote_MNh) hourly data is caused by the drastic misfit of values while the wall gets no direct insolation, before 13:00 o`clock (on 11[th] of August, the middle of the reference period).

Air temperature measured onsite $T$(onsite_uni) is slightly higher compared to $T$(remote_TXL) and $T$(remote_MN) but values are in a close range. This is well in line with studies on the intra-urban variability of $T$ (Eliasson and Svensson, 2003; Fenner





et al., 2014). For daily resolution, $T$(remote_TXL) and $T$(remote_MN) seem to be a sufficient approximation for onsite measured data with R² being 0.62 for both remote_TXL and remote_MN. For even better approximation, an empirical factor for calculating $T$(onsite) from $T$(remote) could either be derived from onsite calibration measurements, or be estimated from

temperature maps, that many cities have commissioned in the past decades, e.g., the environmental atlas of Berlin (Geoportal Berlin, 2001).

Values of $rH$(onsite_uni, onsite_height) are slightly lower compared to $rH$(remote_TXL) and $rH$(remote_MN). Remote and interpolated remote data seem to be nonetheless a good approximation for onsite measured data (R² = 0.42 for remote_TXL and R² = 0.43 for remote_MN). Intra-urban variation in humidity was found to be linked to urban surfaces and forms, such as

imperviousness, urban density, and vegetation abundance (Cuadrat et al., 2015; Wang et al., 2017; Yang et al., 2017). Humidity data from remote climate stations can be used to estimate humidity at an urban façade, but the surroundings of the climate station should be comparably urban and impervious. In most cases, climate stations are rather not influenced by urban surroundings.

Regarding monitoring and irrigation planning, measuring $T$ and $rH$ directly at the greened wall would be the best strategy,

especially as combined $rH, T$ sensors are comparably cheap. Given the deviations regarding $T$ and $rH$, also daily $VPD$(remote) differs clearly from $VPD$(onsite) by 10 % to 31 % for data from TXL (R² = 0.99) and MN (R² = 0.97), respectively (Fig. 4). For hourly resolution, the pattern of the study site to be drier and warmer applies as well, the deviations between $VPD$(remote) and $VPD$(onsite) increase to 23 % (remote_TXL, R² = 0.74) and 39 % (remote_MN, R² = 0.65).


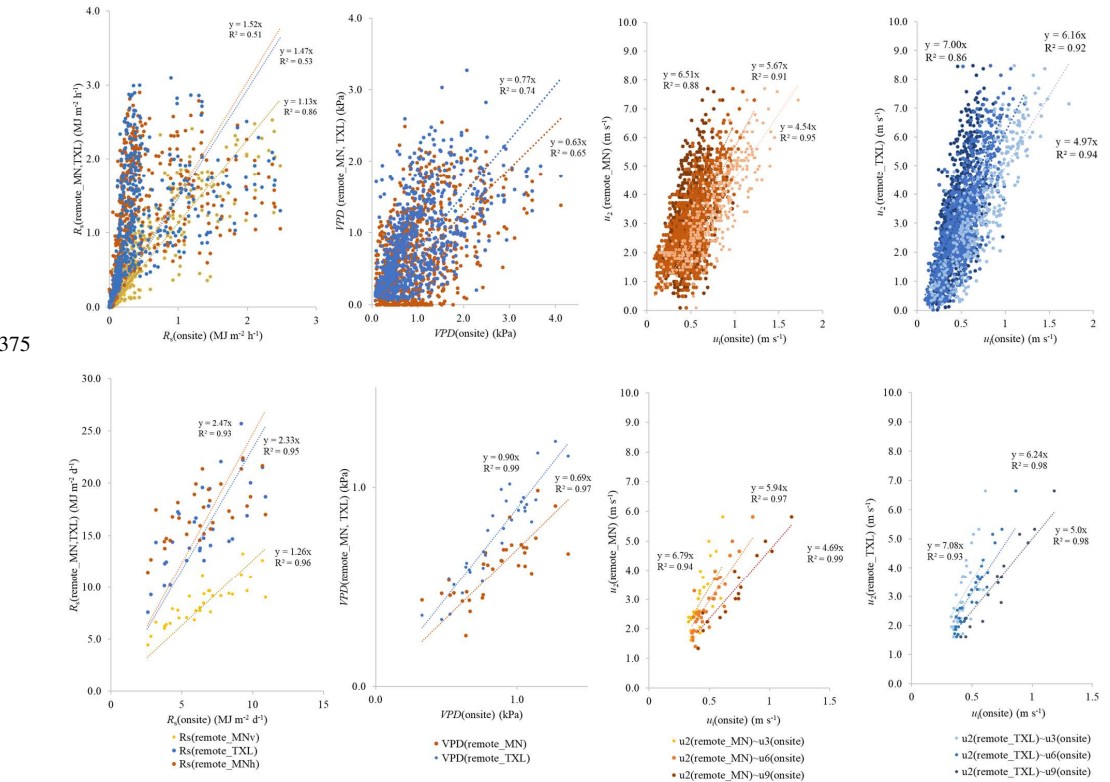

**Figure 4:** Left and middle left: shortwave radiation ($R_S$) and vapor pressure deficit ($VPD$) from remote climate station Tegel (TXL, blue) and Meteonorm (horizontal plane: red, vertical plane: yellow) versus onsite measured data; Middle right and right: Wind speed from TXL and MN versus onsite data measured at three different heights (dark: 3m, medium: 6m, light: 9m); please note that *data*(remote_TXL) and





*data*(remote_MNh) were measured on a horizontal plane, while onsite measured data and *data*(remote_MNv) refer to the vertical plane. (25/07 – 29/08/2014, n = 864, respectively 36).

Compared to $u_3$, $u_6$, $u_9$(onsite) measured in 3, 6 and 9 m height, $u_2$(remote_TXL) and $u_2$(remote_MN) measured and interpolated from stations with open surroundings are 5 to 7 respectively 4.5 to 6.5 times higher (Fig.4). Values for $R^2$ indicate a very high predictability of onsite measured values from remote station data.

Because wind flow in urban areas is shaped by local conditions and obstacles (Barlow and Drew, 2015), high-resolution numerical simulations based on computational fluid dynamics are needed but are complex and computationally expensive (Murakami et al., 1999; Di Sabatino et al., 2013).

Mean wind direction at TXL was orthogonal to the canyon at the study site in 64 % and parallel to it in 36 % of the investigated days. According to Christen (2005) and Eliasson et al. (2006), the relationship between above-roof and intra-

canyon wind speeds changes with above-roof wind direction. Yet, by splitting the time series according to these two cases of wind direction and calculating separate sets of factors, no significant differences could be found for this study. Accordingly, no improvement in model accuracy was achieved by including wind direction, which consequently was not further considered in calculations. The observed independence from wind direction might be due to the heterogeneous surroundings of the study site (multiple large building of varying height and a large avenue of 115 m width) and associated channelling

effects (Oke et al., 2017).

The wind did have a relevant impact on ET in its respective range of values although that was not represented by a simple linear regression line as for $R_S$ and $VPD$ (Fig. 3). There are, however, dependencies visible for the lowest wind speeds and the highest wind speeds. In a global sensitivity analysis, interaction effects between the variables became visible, especially for wind speed (Saad 2020). Wind flow in urban canyons is highly turbulent and therefore difficult to simulate on a physical

basis. However, on a daily scale it was shown that it can be approximated on an empirical basis with sufficient accuracy. It can be assumed that the approximation is more accurate in higher levels of the canyon, because closer to the ground the wind flow is less dependent on large-scale wind dynamics and more on specific local conditions and obstacles. This is in line with the observations of Nakamura and Oke (1988) of high scatter at low wind speeds due to a decoupling of the flows inside and above the urban canopy. Wind measurements in at least three different heights in the canyon are necessary for calibration of

the empirical factors. Ideally, the instruments should be placed at least one meter in front of the facade to avoid disturbances. Additional wind speed measurements above roof-height could give valuable insights in the local wind field and facilitate a partitioning of the empirical factors into intra- and above-canopy effects. Furthermore, variations in wind speed profiles along the longitudinal canyon axis have been observed due to increased levels of turbulence at intersections and corners (Carpentieri et al., 2009; Oke et al., 2017), so that the transferability of measurements at one facade to other facades even

within the same canyon might be restricted.

The missing comparability between the results of sensitivity analyses in different studies underlines the importance of a thorough and location-specific analysis of insecurities of models based on the Penman-Monteith equation. Researchers often need to decide on which climate variables to focus and quantify with costly measurements and which climate variables to estimate from existing data. A sensitivity analysis of the intended model with variables in their location-specific range of

values can help with that decision.

### 4.3 Comparison of the verticalization approaches for $ET_c^{vert}$

*ET* of the green wall at the test site calculated with onsite or remote meteorological data both for the horizontal and vertical surface is compared to the lysimeter measured *ET*(lysi). All calculated *ET* data refers to a plant-covered wall area of 18 m²

and considers a crop coefficient of $K_C = 1.21$.





The importance of verticalizing input data is most apparent when comparing total sums of modelled $ET$ and measured lysimeter data ($ET$(lysi)) during the study period (25/07/–29/08/2014). Total measured $ET$(lysi) in the period has been 1317.9 L. Without verticalization, ET is overestimated by +108 % ($ET$(remote_TXL)) or +100 % ($ET$(remote_MNh)). Verticalization of remote data can lead to less overestimations of +24 % ($ET$(remote_MNv)). Best estimation of total $ET$

could be achieved using verticalized onsite data both uniform and height-adapted (underestimations of $ET$(onsite_height) being -4 % and $ET$(onsite_uni) being –11 %).

Best performance on the daily and hourly basis was expected for the verticalized, height-adapted dataset. In this specific case, the difference using height adapted data - $ET$(onsite_height) versus $ET$(onsite_uni) - is visible, though it might not be relevant for the irrigation in practice. Daily courses of $ET$(onsite_uni) and $ET$(onsite_height) (Figure 6) show only small

differences between the two datasets. That is due to the relatively homogeneous insolation of the test wall. Neighboring buildings or other obstacles have a low impact on the insolation of the test wall – specifically during the reference period. In north summer, due to the high altitude of the sun the shading effect is not as drastic as in spring and autumn. For situations with closer and/or higher buildings, the insolation would be more heterogenous on the wall. As the two datasets are quite similar at the study site, the validity of the model is satisfying for both $ET$(onsite_height) and $ET$(onsite_uni).

Using hourly data resolution and onsite measured data, the validity for the model for the study site using height-variable insolation data, $ET_c^{vert}$(onsite_height), is comparable (RMSE of 1.2, MAE of 0.73) to the uniform scenario $ET_c^{vert}$(onsite_uni) (RMSE of 1.21, MAE of 0.74). With a difference of +1 %, the height-adapted model is more accurate than the uniform model (-5 %). For daily resolution, $ET_c^{vert}$(onsite_height) underestimated the measured data with only -2 % while $ET_c^{vert}$(onsite_uni) underestimated the measured $ET$(lysi) by -9 %. Thereby, with 0.95 and 0.94 the coefficients of

determination R² for daily resolution are higher for $ET_0^{vert}$(onsite_height) and $ET_0^{vert}$(onsite_uni). For hourly data, coefficients of determination are 0.76 for $ET$(onsite_height) and 0.74 for $ET$(onsite_uni). Applying remote data resulted in high overestimations with comparable high coefficients of determination. When using remote climate station data either directly or interpolated, average overestimations between + 27 %, + 89 %, and + 99 % occur in comparison to - 5 % and + 1 % when using uniform or height-dependent onsite data, respectively (Fig. 5). The height-dependent insolation should be

applied whenever possible, especially for walls showing heterogenous insolation patterns – or in other words a shadow on the wall.






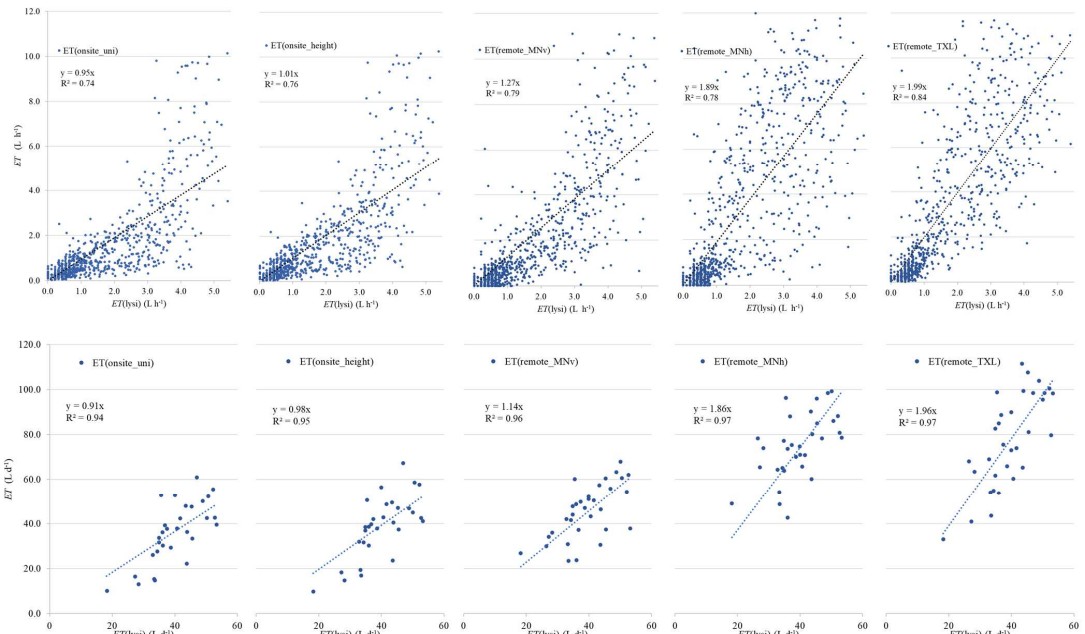

**Figure 5:** Hourly (top, n=864) and daily (bottom, n=36) $ET$ of the green wall for 25/07/2014 – 29/08/2014: Comparison between measured lysimeter data $ET$(lysi) and calculated (from left to right) $ET_c^{vert}$(onsite_uni), $ET_c^{vert}$(onsite_height), $ET_c^{vert}$(remote_MNv) (interpolated, standard data but verticalized radiation input), $ET_c^{vert}$(remote_MNh) (interpolated, standard data), $ET_c^{vert}$(remote_TXL) (remote single station data, standard meteorological data). A crop coefficient of $K_C = 1.21$ has been used for calculating $ET_c$. The greened/covered wall area was 18 m².

Using remote and non-verticalized data overestimates $ET$ during the day for two reasons. First, $ET$ is generally estimated too high. Second, as the study wall is west-oriented, the wall is exposed to direct sunlight only starting from noon. The non-verticalized data does not take this into account as can be seen best for the cloudless day (see 06/08/ in Figure 6).

During the night, calculated $ET_c^{vert}$ underestimates measured $ET$(lysi). This night-time phenomenon could be caused by the measuring concept at the study site: as $ET$ is measured as the water uptake by both plants and soil from the weighed water reservoir. Thus, water uptake might be misinterpreted as $ET$. For example, when the plant replenishes water, after a high-transpiration day. As this takes place at night it was visible in $ET$(lysi) but not accounted for in the $ET_c^{vert}$ calculations. The absorbed water could then be transpired during the day without causing further (visible) water uptake from the lysimeter. Such behaviour is well known from trees as stem water refill and storage (Čermák et al., 2007, Raven et al., 2013). The course of daily $ET$(lysi) would thus appear smoothed compared to the peaks of e.g. $ET_c^{vert}$(onsite_height). Another reason could be that at the site, the building emits long-wave radiation during the night, leading to increased cuticular $ET$ (Hoelscher et al., 2018). The night-time temperature differences of wall surface and air can be as high as 2.5-3 °C (Hoelscher et al., 2016).

According to Allen et al. (1998), the relative contribution of evaporation $E$ from the soil surface to evapotranspiration ET is never less than 10 % during the vegetation period for horizontal crops. For climbing plants, the $ET_0$ formula will thus overestimate $E$ as leaf area to soil surface ratio is much higher than for grass. Schwarzer et al. (2015) measured evaporation and transpiration of a VGS at the same study site in Berlin. They showed evaporation from the 1 m² container accounted for approx. 2 % of total water uptake of *Fallopia baldschuanica*. For wall-based systems, the ratio could be higher.





However, for the investigated case, adjusting $K_C$ (Allen et al, 1998) was an effective measure to reduce differences between calculated $ET$(onsite_uni, onsite_height) and measured $ET$(lysi).

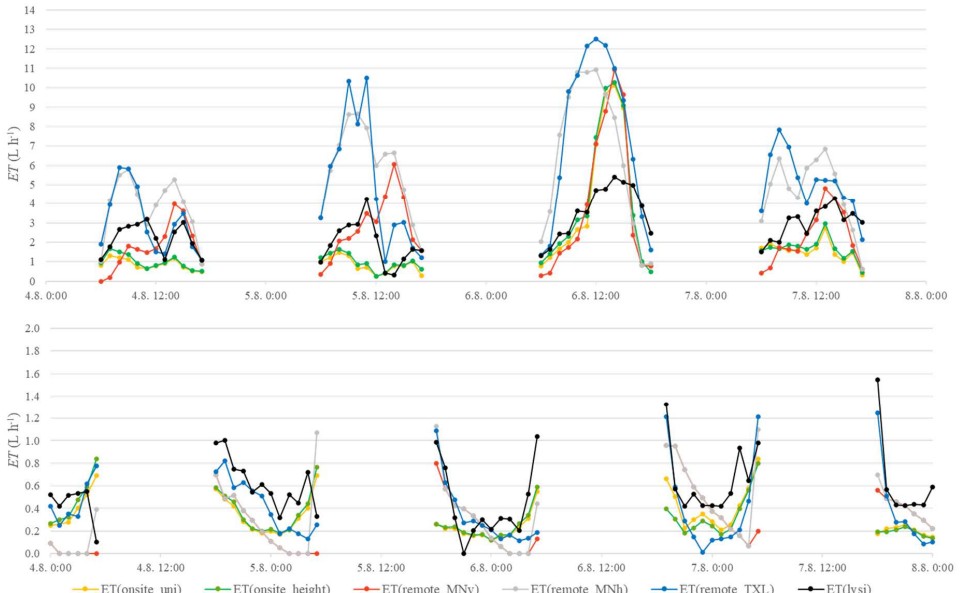

**Figure 6:** Time series of hourly $ET$ (04/08/-08/08/2014) during daytime (top) and nighttime (bottom) calculated from onsite data without height dependency (onsite_uni), from onsite data with height dependency (onsite_height), from remote climate station data interpolated to the site and referring to vertical plane (remote_MNv), from the same interpolated data but referring to the horizontal plane (remote_MNh), from remote data (remote_TXL), and measured lysimeter data $ET$(lysi); all $ET$ values refer to 1 m² of vertical area; All calculated $ET$ data include the crop coefficient $K_C$ = 1.21; note that the accuracy of $ET$(lysi) is 0.01 L m$^{-2}$.

Some limitations of the present study need to be highlighted, regarding both the measured and the modelled $ET$ data. Next to the previously mentioned time lag between water uptake and transpiration, the possible limitation of hydraulic flow in tall plants (Ryan and Yoder, 1997) might impede the accuracy of the lysimeter data. Additionally, even though care has been taken in the experimental design to achieve the closest possible alignment with $ET_0$ (sufficient water supply through hydroponic system), the lysimeter measures actual $ET$, which limits comparability to $ET_0$ calculated with the Penman-Monteith equation.

Regarding the modelled data, uncertainties exist linked to the original conception of the Penman-Monteith equation for agricultural crops. These dense crops form homogenous canopies, which is reflected in the single-layer approach of the Penman-Monteith equation treating a canopy as one big leaf (Allen, 2005). VGS, on the contrary, are isolated plants and therefore have, in relation to their mass, a much larger surface area exposed to the influences of their environment (e.g., advection) as observed for isolated plants by e.g., Rana et al. (2019). Their vertical orientation has impacts on convective heat losses (Gunawardena et al., 2017), radiation distribution in the plant canopy (Jones, 2013), storage and release of heat from the wall (Hoffmann, 2019) and wind loads experienced by the plants (Aly et al., 2013). The major advantages of using the standardized Penman-Monteith equation, even if it might not be perfectly suited for VGS, are the great comparability of its results and the possibility to empirically adjust it with a crop factor if plant-specific parameters like stomatal resistance are unknown. The seasonal phenology of the climbers and its impact on $ET$ calculation should be further investigated. When modelling the $ET$ of vertical green over the course of a year, a phenology module considering yearly climatic variations, urbanization effects (Neil and Wu, 2006) and species-specific leafing progression should be included.



**5 Conclusions**

In this study we propose a model calculating $ET_0$ from green walls $ET_0^{vert}$ and tested it with a series of onsite and remote
climate data. Although VGS and grass are certainly different in terms of morphology and physiology, $ET_0$ and $K_C$ can be
used to predict the water demand of VGS.

Using onsite measured input data, an uncertainty of less than 10 % can be achieved for hourly ($R^2 > 0.74$) and daily ($R^2 >$
0.94) data. Considering height-dependencies of radiation and wind speed, accuracies get even higher. If only remote climate
station data is available, the verticalization of radiation data is highly effective. For interpolated data, the verticalization
could reduce the uncertainty from 86 % to 14 % overestimation. The simple use of horizontal remote station data seems not
to be advisable as it resulted in overestimations of almost 100 %.

Deviations between calculated and measured hourly courses of $ET_c^{vert}$ suggest using both more sophisticated lysimetry
which enables the separation of plant water uptake and $ET$ and adapted modelling approaches, e.g., including dynamics of
stomatal conductance. However, for irrigation planning on a daily base, this is not necessary.


**Author contribution**

K. Hoffmann: Conceptualization, Methodology, Software, Validation, Formal analysis, Investigation, Data curation, Writing
- Original draft, Writing - Review and Editing, Visualization, Project Administration;

R. Saad: Conceptualization, Data curation, Methodology, Investigation, Software, Writing - Original draft, Writing - Review
and Editing;

B. Kluge: Conceptualization, Methodology, Software, Investigation, Data curation, Writing - Original Draft, Writing -
Review and Editing;

T. Nehls: Conceptualization, Methodology, Software, Validation, Formal analysis, Investigation, Writing - Original draft,
Writing - Review and Editing, Visualization, Supervision, Project Administration, Funding Acquisition;


**Competing Interests**

The authors declare that they have no conflicts of interests.

**Acknowledgements**

This study reflects the results of a comparable long development of ideas, measurement concepts and model approaches during
four different projects. The authors are grateful to DFG for funding the Research Unit 1736 "Urban Climate and Heat Stress",
to the German Federal Ministry of Education and Research BMBF for funding "Vertical Green 2.0" (FKZ: 01LF1803A), "Blue
Green Streets" (FKZ: 033W103G), and the Federal Ministry for Economic Affairs and Climate Action, BMWK for funding
"U-green - building physical assessment of facade greening and green roofs" (FKZ: 03EN1045C) and the Technische
Universität Berlin for funding the Center for Innovation and Science on Building greening.

**Abbreviations and symbols**

| | |
|---|---|
| $\gamma$ | psychrometric constant (kPa °C$^{-1}$) |
| $\Delta$ | slope of the saturation vapor pressure-temperature curve (kPa °C$^{-1}$) |
| $C_d$ | denominator constant; for short surfaces and hourly timesteps equals 0.24 during daytime and 0.96 during nighttime (s m$^{-1}$) |
| CFD | Computational Fluid Dynamics |
| 540    $C_n$ | numerator constant; for short surfaces and hourly timesteps equals 37 (K mm s$^3$ Mg$^{-1}$ h$^{-1}$) |





|  |  |  |
|---|---|---|
| | DWD | German Weather Service "Deutscher Wetterdienst" |
| | $E$ | Evaporation (L h$^{-1}$, mm h$^{-1}$ or L d$^{-1}$, mm d$^{-1}$) |
| | $ET$ | Evapotranspiration (L h$^{-1}$, mm h$^{-1}$ or L d$^{-1}$, mm d$^{-1}$) |
| | $ET_0$ | Horizontal Grass Reference Evapotranspiration calculated according to Allen et al. (1998) for short surfaces (L h$^{-1}$, mm h$^{-1}$ or L d$^{-1}$, mm d$^{-1}$) |
| 545 | $ET_0^{vert}$ | Verticalized $ET_0$ calculated according to Allen et al. (1998) (L h$^{-1}$, mm h$^{-1}$ or L d$^{-1}$, mm d$^{-1}$) |
| | $ET_0^{VGS}$ | Verticalized $ET_0$ of a full stand VGS (L h$^{-1}$, mm h$^{-1}$ or L d$^{-1}$, mm d$^{-1}$) |
| | $ET_C^{vert}$ | Verticalized crop evapotranspiration, i.e. $ET_0$ multiplied with crop coefficient $K_C$ (L h$^{-1}$, mm h$^{-1}$ or L d$^{-1}$, mm d$^{-1}$) |
| | $ET_P$ | Potential Evapotranspiration according to Allen et al. (1998) (L h$^{-1}$, mm h$^{-1}$ or L d$^{-1}$, mm d$^{-1}$) |
| | $e_s$ | saturation vapor pressure at 1.5–2.5 m height (kPa), calculated as the average of saturation vapor pressure at max. and min. air temperature |
| 550 | $e_a$ | mean actual vapor pressure at 1.5–2.5 m height (kPa) |
| | FAO | Food and Agriculture Organization by the United Nations |
| | G | soil heat flux density at the soil surface (MJ m$^{-2}$ h$^{-1}$ or MJ m$^{-2}$ d$^{-1}$) |
| | h | height (m) |
| | h | horizontal |
| 555 | $K_C$ | crop coefficient |
| | $k_u$ | height-dependent wind factors |
| | MAE | Mean absolute error |
| | MN | Meteonorm |
| | n | number of data points |
| 560 | R² | Coefficient of determination (-) |
| | Ray | RayMan model was employed (Matzarakis et al., 2007, 2010) |
| | RMSE | Rooted mean square error |
| | $R_n$ | Net radiation as difference between incoming net shortwave and outgoing net longwave radiation |
| | (MJ m$^{-2}$ d$^{-1}$ or MJ m$^{-2}$ h$^{-1}$)$R_s$ | Solar global radiation (MJ m$^{-2}$ d$^{-1}$ or MJ m$^{-2}$ h$^{-1}$) |
| 565 | SEBE | Solar Energy on Building Envelopes (SEBE) model (Lindberg et al., 2015, 2018) |
| | UHI | Urban Heat Island |
| | $u$ | wind speed (m s$^{-1}$) |
| | $u_i$ | wind speed in height i (m s$^{-1}$) |
| | UTC | Coordinated Universal Time |
| 570 | $T$ | air temperature (°C) |
| | TXL | German Weather Service station Berlin-Tegel |
| | v | vertical |
| | VGS | Vertical greening system |
| | $VPD$ | Water vapour pressure deficit |
| 575 | | |

brackets denominating data sets as in $ET_0^{vert}$(onsite_height)

|  |  |
|---|---|
| (lysi) | data set on water consumption of the VGS measured with the lysimeter |
| (onsite_uni) | onsite measured data set from one height, uniformly applied for the whole facade height |
| (onsite_height) | onsite measured or derived data set for the different height increments of the facade |
| 580 (remote_MNv) | data set from the Meteonorm database, radiation data verticalized according to Perez et al. (1991) |
| (remote_MNh) | data set from the Meteonorm database, all data from standardized (horizontal) stations |
| (remote_TXL) | data set from the DWD Station at the former Airfield Berlin-Tegel (IATA airport code: TXL, RIP!) |

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
