# Peer review of "Modelling reference evapotranspiration of green walls $(ET_0^{vert})$"

_Hydrology and Earth System Sciences, 2023_

## Author Comment (AC1)

Manuscript ID: hess-2023-221

Reply to anonymous Reviewer #1

**RC1.2: My overall impression of the article is that it feels a bit shallow, especially in the methodology.**

Your feeling might not be meant like this but we receive "a bit shallow" in fact as a compliment. It was a bit surprising to see that nobody has tried the obvious before. Instead, we have found only case study data, which is not transferable. In our opinion, it is the strength of this research that we have adapted the well-established, well-acknowledged and international applicable $ET_0$ approach based on the process-based Penman-Monteith approach to the vertical in an urban street canyon and validated it with experimental data. However, measuring water consumption of vertical green was all but simple or "shallow". While in lines 245-267 we describe the measuring setup, we agree to the reviewer that this description needs to be improved by a technical drawing.

**RC1.3: The Introduction reads more like a review article, being quite extensive in digging up older studies on the subject of ET and vertical walls, and the relation to localised data usage. However, I was struggling to see exactly what the point is the authors are trying to make here with this extensive literature research. The subsection on Measuring ET just lists several ET values from different studies - is this really necessary? It would be much more easily accessible if it were tabulated for instance so the Introduction can be made much more concise and to the point. The authors don't reach their objective of the study until page 5, at which point it was still fairly unclear to me what the actual point of the paper is, how the authors will tackle the issue, exactly what the issue is etc.**

Thank you for your comment. There are several personal preferences about how introduction sections should be written. How ever, we will shorten the introduction accordingly, will present data on ET of different studies in a table as proposed and will name the aims of our study in the beginning and deduce it from the state of the art afterwards.

**RC1.4: In the Theory section, one particular sentence that struck me as odd was in line 220-222, mentioning that VPD is highly influential on ET, is height-dependent on wind speed, but that would require CFD or local measurements so the authors just omit it...? I find it hard to justify this choice given that the authors explicitly state its importance on ET only to then ignore the variations.**

With this criticism both reviewers are right! We agree that this paragraph needs clarification and will be revised. Of course, VPD has a high impact on ET (as shown in Figure 3), high variations of VPD lead to high variations of ET. However, the temperature differences at one point of the facade in the diurnal course are much higher than the differences in the height profile, as shown in Offerle et al. (2007), see the figure below. A height dependency is only visible for short periods. Therefore, we discuss that the impact of the height profile compared to a uniform VPD is neglectable.

**Fig. 4** Air temperatures in the canyon as a mean difference from R4 for sunny days in each subset as in Fig. 2: (**a**) summer, (**b**) spring and autumn, (**c**) winter. The line marks the canyon top. All air thermocouples were used to calculate the mean vertical temperature profile i.e. horizontal variation within the canyon is ignored

[Figure]

**RC1.5: The Material/Methods section then feels lacking in some details: the scientific quality of the article would benefit some more detailed description or visualization of the measurement setups (both at the experimental site and the external measurement sites). Especially the location of the sensors at the experimental site are important, given the strong non-linearity of environmental properties in the urban area, with wind speed as the most obvious one. The authors very briefly touch upon wind influence but seem to conclude that it is not important for ET estimation - this might also have to do with the location of the wind measurements and the behaviour of the wind at its microclimate.**

It is a bit difficult to discuss feelings. From our point of view, the M&M was complete with the reference to Hoelscher et al. (2018), where the setup is described in detail. However, we took your comment serious and as an occasion to re-write the M&M section in a way, that the article is self-explaining. We will add a technical drawing of the measuring setup, a re-draw of the figure 1 from Hölscher et al., 2018 and we will combine it with a technical detail drawing of the water consumption measurements mentioned above.

[Figure]

Fig. 1. Design of lysimeters for the outdoor experiment: transpiration measured by sap flow sensors and balance: a) sap flow sensor with radiation shield (for 5 plant stems in total), b) balance measuring the water uptake for all 32 stems, c) climbing plants (*Fallopia baldschuanica*), d) rooting zone in the hydroponic system, e) free outflow, f) intake, g) water reservoir refilled automatically, h) constant water level, i) data logger, j) pump, k) cover.

**RC1.6: Another example of missing clarity of the interpolation of climatological data from the of-site stations - how is this done? Is the landscape taken into account at all? What method of interpolation has been used, has this been verified, is it susceptible to errors etc etc. These details are crucial since they can influence the validity of the conclusions.**

We fully agree with your comment, interpolating meteorological data is a book on its own. As interpolation of meteorological data is beyond our competences, we decided to check the usability of available data sets. So, we used interpolated data delivered by MeteoTest (as described in lines 279 ff). Regarding the interpolation method, we need to refer to Meteotest (2020), as done in the manuscript. We will add some details about the calculation of incoming solar radiation on a vertical pane.

**RC1.7: The Results section has a lot of difficult to read text, formatted with Rs(remote_TXL) etc etc, with all the parentheses and mentioned values makes the text a slog to get through. I'd advise the authors to see if these cannot be summarized in a table as well, and to really get to the focus of the results.**

Thank you for your comment. Indeed, the descriptions given in lines 294 ff could be substituted by a table making our nomenclature more accessible. So we will rewrite that section.

**RC1.8: The figures themselves are also quite hard to read, especially in print.**

Thank you for this feedback. We agree that the overall font size in Copernicus' text template is already quite small. The same applies for the figures. We will increase the font size, enhance contrast and resolution for the figures. We will aim to change figures 4,5, and 6 to be full page figures.

**RC1.9: The results also feel like they don't quite go deep enough in explaining things, with some statements that feel fairly obvious (e.g. radiation values being of because the wall is shaded for a part of the day - something very obvious that you could have corrected for beforehand). So most of the results feel like quite generic statements, even though there are very interesting tidbits of knowledge in there that could really help judging the quality of using off-site data or not. For instance the possible influence of night-time longwave radiation on night-time ET - interesting stuff to explore.**

Thank you for your recommendation. We agree that our data set is really interesting for more pronounced process studies. E.g. the mentioned "**night-time longwave radiation on night-time ET - interesting stuff to explore**" have already been investigated. The pronounced ET during nighttime lead to a new base line correction for sapflow measurements, see Hölscher et al.(2018) – we will point to that fact accordingly.

However, the focus of this article is to develop and to evaluate the adapted standard evapotranspiration $ET_0$ approach using different data sets with experimental data. Our aim is to adapt the model to use available meteo data for sites in urban settings. Your statement "**the wall is shaded for a part of the day - something very obvious that you could have corrected for beforehand**" is confusing. It seems that our calculation concept has not been understood, which points to possible weaknesses in the text. We will check it accordingly with special care and will explicitly state that in the concept section.

**RC1.10**: **In summary, the structure of the text and figures as well as the phrasing of the aim and results of the paper could use some serious work, but with some more in-depth information on the setup, as well as a more in-depth look at the actual results, would result in quite an interesting paper.**

Again, thank you very much for your dedicated work on the review and your appreciation of the overall objectives of the manuscript. We are willing to revise the manuscript accordingly and thoroughly.

Thomas Nehls on behalf of the authors

---

## Author Comment (AC2)

Reply to anonymous Reviewer #2

**General:**

**The manuscript is presenting a modelling concept for evapotranspiration determination of green walls, which is a relevant topic, specifically nowadays when greening infrastructure is a highly recommended measure to reduce urban heat and prevent drought. However, to the reviewer's opinion, the method chosen is inappropriate due to violation of underlying (homogeneity) assumptions in the reference method chosen.**

We thank you for your thorough and dedicated review and are happy that you appreciate the relevance of the presented topic. We disagree on your opinion on the violation of the homogeneity assumption and will discuss that in detail below.

**Selecting (much) simpler approaches (e.g. Priestley-Taylor) might be more suitable and should be exploited.**

In our view, the process-based Penman-Monteith is the more reliable approach for this purpose. Using simplified approaches such as Priestley-Taylor would require site specific empirical factors to the climatic conditions on site. It was explicitly the aim of this study to test a process-based approach that can be transferred to other sites and that can be used with full sets of onsite data or remote data with the correspondingly higher uncertainties.

**An additional analysis, examining the influence of actually received radiation, might provide interesting insights.**

We did this in section 4.2, which's aim it is to show the differences in input parameters between the experimental site and remote stations.

**Reasoning and explanations are often inaccurate/incomplete, possibly due to leaps of thought that are not well explained.**

Thank you for this remark. Also inspired by the remarks of Reviewer #1 we thoroughly revised the manuscript, and we agree that in some parts of the text, further explanations and clarifications would improve it.

**Certain parameters are not taken into account in the analysis simply because they were not available: this is not justifiable reason (and in itself a reason to reject the manuscript).**

Here, of course we disagree. We would like to clarify that not the parameters themselves are omitted, but the assumption of their variability in height (in our experimental setup). This decision is based on the findings of a study by Offerle et al. (2007). We will clarify our argument in the manuscript. Additionally, we would like to add that we indeed have shown that the approach is robust enough to produce results with an uncertainty small enough to be a good base for irrigation demand calculation and for some studies that aim to predict the effectiveness of urban green infrastructure as climate change countermeasures.

**The added abbreviation list is appreciated, but still abbreviations should be explained at first use, to facilitate easy reading.**

That will be changed in the text accordingly.

**English here and there might need slight polishing.**

Thanks for this remark. A proof reading has been done before submitting but will be done again after revision of the text.

**Summarizing I recommend either reject (and submit to other journal) or major revision. The above comments are detailed below:**

We agree to thoroughly revise the manuscript. Of course, the article deals with (i) blue-green infrastructure aspects and (ii) the "verticalization" of $ET_0$, a topic of importance for urban ecohydrology. Therefore, we have chosen HESS and trusted the editorial board when they accepted the article to be reviewed by HESS. Because of the preprint-publication in HESS and the reactions of the scientific society (recommendations and reads), we do not favor a withdrawal and submission to another journal. However, that decision is obviously up to the editorial board.

**Text:**

**RC2.2: Page 2, line 42: Remove "on"**

Thanks. Will be changed accordingly.

**RC2.3: Page 3, line 56: "contributions of 13% up to 73%" ; reviewer assumes these percentages are the relative contribution to the total cooling effect. If yes, this should be explained more clear. In addition the total amount of cooling should be specified in one way or another to make more clear what are the quantitative effects.**

In this paragraph, we are referring to a previous work done by Hoelscher et al. (2016). We will explain that in more detail as requested.

**RC2.4: Page 3, line 85: "results cannot be generalized": it is unclear why not. Results can just be compared to nearby standard meteo station data as is the case with other ET observations; They often also do not show a straightforward relation to the ET observations in a standard meteo station, but that is not a reason not to use standard meteo station data for generalization.**

Standard meteo-stations according to WMO standards are set up in a surrounding differing a lot from the given urban setting at a building's façade. In the urban setting, objects in the direct vicinity influence the input parameters. Therefore, similarity between measured data at the façade and standard meteo station data is limited and that is one question of this study.

**RC2.5: Page 3, line 90: Replace "with" by "where" and replace "delivering" by "delivered the"**

Thanks. Will be changed accordingly.

**RC2.6: Page 4, line 94: Replace "included" by "used"**

Thanks. Will be changed accordingly.

**RC2.7: Page 3, line 93: Unclear which validation is meant here. Also unclear to what the correlation refers (correlation between what?).**

and

**RC2.8: Page 4, line 97/98: Unclear why surface temperature is just a proxy for ET; please explain more.**

We agree, that need to be discussed in more detail, especially what are the shortcomings of the study.

**RC2.9: Page 4, line 101: Unclear to which regression the correlation figures refer and unclear what are the "panel system" and the "planter box system"; please explain further.**

We agree, it will be better described in the revised version.

**RC2.10: Page 5, line 136: "To refine ….in the model": Unclear what is meant here; please explain further.**

Thanks for your hints. We will explain the referenced literature more in detail.

**Section 2. "Theory"**

**RC2.11: Page 6, line 179: "all these parameters need to be verticalized". Reviewer would like to remark that other plants, also in a non-urban area, have vertical dimensions and, especially in a non-homogeneous environment, may or may not experience sensitivity to the parameters mentioned (Rn, Rs, u, T, VPD) in a manner that is not independent of height**

We agree to the reviewer's remark. The difference between - let's say maize or bushes or trees - is that their leaves are irradiated the whole day long and are not shaded systematically like in the case of facades (and if they have partly shaded leaves, there will be others in the full sun then). It makes a difference that a tree has a round footprint and a façade greenery will never see the sun from the backside. We agree, that once the reviewer remarks that, we have to say that explicitly in the manuscript, to make the difference between a bush, a tree, a wheat field and a façade greenery clearer, especially as later the discussion about the homogeneity of the evaporating pane has to be done.

**RC2.12: Whereas it is obvious that the influence of mentioned parameters is varying with height in such circumstances, employing (simple linear) regressions to correct for this (multiple) influences is violating the original homogeneity assumptions underlying the PM approach (so-called "big leave approach"), reason for which it would be more just to employ such regressions to much simpler approaches, or simply to standard met-station observations of ET (when fully 3-D models, which in principle are required for conditions like the present, are not an option).**

We disagree. For us it gets not clear, why the height dependency should not be corrected by linear factors, if meteorologists found the wind to slow down over the roof with increasing canyon depths. It is the concept to assume that the planar façade greenery behaves like a set of homogenous vertical height-depending increments. We might have not made that clear enough. We will add a conceptual figure to figure 2 to make that point clearer. Whether a full 3 D model is needed or not can be discussed in other studies. In fact, such models are not available for the majority of urban sites. To our opinion it gets not clear from the reviewer's statement, why we should use simple ET observations of met stations when we can calculate $ET_0$. It is not the aim to calculate actual ET for the façade but $ET_0$ as it is the need to predict long term water demand for façade greeneries in order to be able to check for sufficient water supply (from rainwater or from greywater, see Pearlmutter et al 2021: Water 2021, 13, 2165. https://doi.org/10.3390/w13162165) Again, if that was not clear from the text we will revise it accordingly.

**RC2.13: Page 13, line 412: "based on the PM equation": as mentioned previously; the underlying assumptions of PM are violated, instead a 3D model should in principle be used. If that is not possible/desirable, a simple regression of ET meteo station data with onsite data and some parameters (radiation being the most important one) would be a more logical approach.**

The reviewer is not right in his "violation" remark, therefore we need to provide clarification about the fact that with the verticalization approach we indeed leave the (one) big leaf approach and instead assume homogeneity for each of the 12 height levels (i.e., 12 big leaves). In our case study, the 12 m height might not lead to extreme differences between the base and the top of the plant. Buit this approach might be more important for higher buildings such as the CDL's(City developments limited) "Tree House project" in central Singapore.

Our approach is still process-based whereas applying simple regressions with standard meteo station data as suggested would lead empirical models which are not transferable to other sites. Using such regressions would necessitate onsite calibration of the according models. Such would be of limited value in times of climate change.

**RC2.14: Moreover, every individual site would require on-site calibration (which will depend on its surroundings), making the approach unsuitable for "universal" application.**

We disagree with this conclusion. Adjusting the input parameters as described, makes onsite calibration unnecessary, apart from adjustment of the dynamic Kc for the individual façade greenery species or even species mixtures.

**RC2.15: In this context one would expect considerations such as relatively homogeneous units (the well-known Local Climate Zone concept by Stewart and Oke (2012) for example) to be applied, or at least mentioned.**

We disagree, as the LCZ concept describes conditions on much broader scale; it was not developed and is not applicable to individual facades in the scale of one to several 10 or 100 m²; The suggested approach to model ET from façade greenery for different LCZ is interesting but has not been our aim and was thus not tested. We want to mention the problem of lacking validation data for such an approach.

We think that we have defended our concept, however, we take your criticism serious and will revise our text accordingly to explain our approach unambiguously/ clearer. The above-mentioned sketch that will complete figure 2 will help.

**RC2.16: Page 7, section "Wall heat flux": it is unacceptable that based on an observation/estimation during summer period the decision is taken that "G might be 0 on a daily basis and during the day"**

The reviewer might actually be right. That estimation will be reassessed based on daily and hourly values for G. We will compare wall heat fluxes and net energy input on a daily and hourly basis and will discuss that accordingly.

**RC2.17: Page 7, lines 220-221: the fact that certain measurements or simulations are not available does not justify simply ignoring them!**

Thank you very much! With this criticism both reviewers are right! We agree that this paragraph needs clarification and will be revised in the following way: Of course, VPD has a high impact on ET (as shown in Figure 3 in the manuscript), meaning that high variations of VPD lead to high variations of ET. However, the temperature differences in the diurnal course are much higher than the differences in the height profile. As shown in Offerle et al. (2007), see the figure below.

[Figure]

**Fig. 4** Air temperatures in the canyon as a mean difference from R4 for sunny days in each subset as in Fig. 2: (a) summer, (b) spring and autumn, (c) winter. The line marks the canyon top. All air thermocouples were used to calculate the mean vertical temperature profile i.e. horizontal variation within the canyon is ignored

**RC2.18: Page 7, line 232: Reference is missing for the "formula" presented. The windspeed profile depends on (effective) surface roughness and atmospheric stability; the approach presented is way too simplistic in an urban area.**

and

**RC2.19: Page 10, line 297: "therefore eq. 4 has been used": No justification is provided.**

Reference for the formula is Allen et al. (1998); we agree that this approach might be too simplistic for the application in the urban environment; However, Allen's ET model makes use of the windspeed measured at the standard-height of 2m. In order to keep that model in its original formulation, we re-calculated the windspeeds measured at 0.15 m from the façade. At first approximation, we used the same formula as Allen et al. (1998) to account for the surface roughness of a grass overgrown wall. We agree that due to complex wind fields in the urban environment, onsite calibration might be needed (see ll. 131-143). On the other hand, $ET_0$ has been successfully used for irrigation planning of different crops applying individual Kc, all violating the 12 cm grass pre-requisite.

**Section 3. "Materials and Methods"**

**RC2.20:  Page 8, line 256: "It"; what is meant here?**

Thanks, will be changed accordingly to "evaporation"

**RC2.21: Page 9, line 265: "Both" is used whereas there are 4 parameters mentioned.**

Thanks. Will be changed accordingly to "u, T, rH and $R_s$ were measured […]."

**RC2.22: Page 9, line 283: Remove "in"**

Thanks. Will be changed accordingly.

**RC2.23: Page 9, line 283/284: "calculated according to Perez et.al., 1991" mentioning only the reference when discussing a crucial calculation is insufficient; provide the equation(s) used and explain what is done. This is also valid for several other parts of the manuscript; it is not clear what is done exactly; see also next remark(s).**

We did not do the calculations by ourselves, here we agree, that the text might be misleading. The calculation of the incoming solar radiation on a vertical plane is provided by the Meteonorm software (and was most probably developed for photovoltaics applications). It follows a geometrical approach (Perez et al., 1991), which is described in detail in the Meteonorm documentation. This model is found to be the best available. For details see: Yang, D. (2016). Solar radiation on inclined surfaces: Corrections and benchmarks. *Solar Energy*, *136*, 288–302. https://doi.org/10.1016/j.solener.2016.06.062). Details on this calculation are not in the scope of this paper nor this journal. We would therefore keep the text as it is here, added by the hint, that Meteonorm actually "did" the calculation.

**RC2.24: Page 9, line 291: "factors ku…were derived": unclear what has happened here exactly (other than a logarithmic model was fitted. More detailed explanation is required.**

We will clarify the sentence accordingly:

*Wind speed u in the height levels 1-12 m were inter and extrapolated from measurements taken at the wall in 3m, 6m, and 9m height. The height-dependency of the wind speed was given by a logarithmic model.*

**RC2.25: Page 9, line 292: "Temperature …height dependent": No justification is provided for this assumption.**

See discussion above (RC2.17)

**RC2.26: Page 10, lines 307-308: "using the mapped horizon": Again; unclear what has been done exactly; more detailed explanation is required.**

We will clarify this sentence accordingly: For the calculation of incoming solar radiation on a vertical plane, the horizon needs to be provided to Meteonorm software; for the mapping of the horizon, see Materials and Methods section (ll. 266-267);

**RC2.27: Page 11, lines 343-354: these figures will vary dramatically from site to site and presumably are driven (only) by shadows received at the urban location ("on-site"); either caused by the immediate surrounding of the site, or by clouds passing by (as is mentioned). To the reviewers opinion an exercise investigating solely the influence of radiation on the regression (using a simple ET-approach, or directly met-station data of ET, instead of Penman-Monteith) would be more useful here.**

Here, we disagree to the reviewer's opinion. We don't see a big advantage in introducing other ET models than the process-based one chosen to be used here. It is the aim of section 4.2 to show the differences in input parameters between an urban site and remote stations. The suggested sensitivity analysis of ET to the input parameters is provided in section 4.3. Testing ET models different from $ET_0$ (FAO) is not the aim of this paper. We acknowledge $ET_0$ still as the benchmark. Testing other models and comparing it to the here introduced approach might be a topic for following studies.

**RC2.28: Page 12, line 367: "climate station": guess this is "meteo station"? if not, mention/define the difference.**

We will change the term accordingly throughout the whole article.

**RC2.29: Page 12, line 370: Replace "comparably" by "relatively"**

Thanks. Will be changed accordingly.

**RC2.30: Page 13, line 400: "However, ….accuracy": add the reference.**

This sentence refers to the afore-mentioned reference (Saad, 2020) which will be clarified accordingly.

**RC2.31: Page 13, line 413: "which climate variable to focus"; as also mentioned by the authors themselves, Rs is the most relevant, especially for green walls.**

We agree to the reviewer. We might have been too conscious with our conclusion. Inspired by the simple generalization by the reviewer, we are drawn to conclude using interpolated data and using verticalized solar radiation

**RC2.32: Page 13, line 420: "Kc = 1.21": from where is this value taken/calculated?**

and

**RC2.33: Page 16, line 477: "Adjusting Kc": unclear how it was adjusted, for which period and on what grounds. Please explain further.**

Information will be added in Materials and Methods, the factor is the result of a parametrization of the model. In the revised version, we will better discriminate between model parametrization (Kc) and simulations using the different remote data sets. We think that this will also make the whole paper a bit clearer.

**RC2.34: Page 14, line 425: Replace "could be" by "is"**

Thanks. This will be changed accordingly.

**RC2.35a: Page 14, lines 431-434: The authors mention here that the variation in received shortwave radiation between the distant and on-site locations is not very drastic in summer (i.e. the period of study) and therefore the relation found is valid. However, they also state that shortwave radiation is the main driver (showing the highest correlation), which would plee (if not demand) for examining this effect in the other seasons as well. In this context instead of "summing" & "verticalizing" the PM approach it would be good to see: What if simply radiation reduction (due to shadow) is taken into account?**

This is exactly what we do when applying the verticalized solar radiation according to Perez except that indirect solar radiation is included. As shortwave radiation is one of the main drivers for ET, we cannot see an advantage of applying a **"simple radiation reduction due to shadow approach"** over the detailed and correct approach of applying the vertical incoming shortwave radiation at the site, considering the possibly very complex horizon due to neighboring buildings, trees etc. with its seasonal dynamics. The fact that in our case in summer, the neighboring buildings do not cast long shadows onto our wall does not mean that simple approaches should be applied in general – as we have discussed. Instead, the incoming shortwave radiation must be measured or simulated as good as possible for simulations of ET 0 vert in different sites all over the world.

**(RC2.35b) What if a very simple ET model/regression (e.g. the Priestley-Taylor approach) is used instead of PM? Especially given the fact that a (rather arbitrarily) Kc coefficient of 1.21 is applied (to reduce differences)**

Thank you for the hint. Priestley -Taylor or any other approach can be tested. However, this was not the aim of our study and we have described our reasons to chose PM over any other modeling approach. We will however take the hint seriously and will cross-check if we have stated that clearly enough. Regarding Kc: it is not something "arbitrary" but - to our understanding - describes the (mainly physiologically and morphologically caused) differences between grass and a given crop.

**RC2.36: Page 15, lines 459-460: "First, ET is generally estimated too high": This is not a reason but an observation.**

This is correct. We change this sentence in the following way: First, ET is generally estimated too high as the energy input from solar radiation is generally higher for the remote, non-verticalized dataset. We would like to draw attention to the following figure showing the differences between the horizontal pane and the vertical on: We will add a similar diagram for the according reference time:

[Figure]

Figure: short wave incoming solar radiation measured onsite on (red) on a horizontal pane and (orange) on a vertical pane. Note that the diffuse radiation.

Please note that a simple reduction would end in a rather not so simple reduction when it is aimed to reach a certain level of accuracy.

**RC2.37: Page 15, lines 462-466: It is unclear to the reviewer what the authors exactly mean here; please provide a more clear description of the processes that occur between the plant, soil and water transport out and into the lysimeter observation.**

To clarify the processes in our lysimeter, we will refer to the Material and Methods section where we state:

- "The water table in the substrate was kept at a constant level by an automatic pump, which replenished the water from a reservoir standing on a balance." (ll. 254-255)

  and

- "Transpiration of *F. baldschuanica* was determined by weighing the mass differences of the water reservoir (Signum1, Sartorius Weighing Technology GmbH, Germany) with a resolution of 2 g min-1. According to Hoelscher et al. (2018), this resulted in an overall accuracy of 0.01 L m-2. Negative differences have been aggregated to hourly and daily transpiration rates (L h-1, L d-1, respectively mm h-1, mm d-1), further denominated as ET(lysi)." (ll. 259-263)

We will revise the M& M section as discussed above and we will add a reference to our Materials and Methods section in the discussion.

**RC2.38: Page 15, line 473: What is meant with "vegetation period"? Please explain, also to clarify the remarks made with respect to soil evaporation versus transpiration rates.**

We agree that this might have been a translation mistake. Here, we refer to Allen's "growing period" between sowing and harvest and applicable as well for deciduous species having a dormancy period. We will change the term accordingly so that the relation to Allen's quantification of soil evaporation vs. transpiration rates gets clearer.

**RC2.39:** **Page 16, line 487: "might impede the accuracy of the lysimeter": Unclear why this would impede the accuracy of the lysimeter; water is still in the plant so it would still be weighed by the lysimeter, is it not? In addition it is not clear how the lysimeter functions when plants are connected to vertical walls; would this not affect the weighing? Please explain n more detail these aspects.**

We will add a more detailed description and a drawing of the experimental setup already given in Hoelscher et al. (2018). We assumed that referencing to this article is sufficient but agree to the reviewer that a more detailed description should be part of this article as well. To answer the question: The water which is taken up by the plant but not transpired cannot be measured by the lysimeter as we only weighted the supply container.

**RC2.40:** **Page 16, line 498: "Penman-Monteith approach", probably Priestley-Taylor approach (which is using radiation/conduction mainly) would be better suited.**

As stated above, we would like to stay focused on PM.

**RC2.41:** **Page 16, line 498: What is meant with "greater comparability"?**

Thanks for the hint, here the terms "applicability and transferability" are more appropriate.

**RC2.42:** **Page 16, line 499: "empirically adjust it with a crop factor"; could also be valid for (all) other approaches to correct/adjust them with a (crop)factor.**

Yes, we agree.

**RC2.43:** **Page 16, line 499: "Stomatal resistance is not (only) a plant-specific parameter.**

Right. It is not only plant specific. Therefore, we did not state that here. We will rephrase to: "species-related".

**Figures**.

**Figure 2: (RC2.44:) It is unclear on which equations the shown profiles are based (especially those for windspeeds and, thus, ET).**

The reviewer is right. The term "expected" is wrong here. Instead "schematic" would have been the right term. From the drawing (for us) it got clear, that the profiles are hypothetical and we used the figure to discuss the concept of verticalization and summing. In the revision we will re-draw the profiles and will give actually measured and calculated height-profiles of input parameters and $ET_0^{vert}$. As mentioned above, the figure will be completed by a conceptional drawing of the "multiple homogenous leaves".

**(RC2.45:) I assume the Rs correction-profile is based on local site DEM specifics, but this remains unclear from the text. Instead of speaking of "verticalization", one could also reason that this is just a manner of "calculating" the correct (i.e. real) amount of Rs that should be input to the PM-equation.**

On the Rs correction: please see ll. 286-289. Of course, "calculating" would be right and sufficient. However, the term "verticalization" describes both reference to vertical surfaces instead of horizontal ones and the consideration of height dependencies. We will therefore prefer to stick to it. To our understanding, "verticalization" means turning the horizontal evapotranspiration pane to the vertical. We will describe that better early in the paper!

**Figure 3: (RC2.46:) In the figure there is only 1 y-axis (showing ET0(remote_TXL) (mm d-1), but the values in the left panel are lower than in the middle and right panel, which is not possible. Middle and right panel (seem to) show similar y-values.**

Thanks for the remark. We will solve this problem. The y-axis is valid for all three plots; however, the x-axis does not show the full range of Rs values (up to 16Wm$^{-2}$ instead of 32 Wm$^{-2}$)

**Figure 6: (RC2.47:) It would be interesting to also plot the net radiation (or shortwave during day and longwave radiation during the night) in the different panels.**

We agree, it would be interesting, and we will add this information as given in the figure above.

Again, we thank you very much for the thoroughly review of our manuscript. We have received very valuable hints that will help us to improve it.

Thomas Nehls on behalf of the authors